# Intranasal Administration of *Codium fragile* Polysaccharide Elicits Anti-Cancer Immunity against Lewis Lung Carcinoma

**DOI:** 10.3390/ijms221910608

**Published:** 2021-09-30

**Authors:** Yuhua Wang, Eun-Koung An, So-Jung Kim, SangGuan You, Jun-O Jin

**Affiliations:** 1Shanghai Key Laboratory of Stomatology, Department of Prosthodontics, Ninth People’s Hospital, College of Stomatology, Shanghai Jiao Tong University School of Medicine, Shanghai 200011, China; heyaniy@hotmail.com; 2Department of Medical Biotechnology, Yeungnam University, Gyeongsan 38541, Korea; eunkoungan@yu.ac.kr (E.-K.A.); sojungkim@yu.ac.kr (S.-J.K.); 3Research Institute of Cell Culture, Yeungnam University, Gyeongsan 38541, Korea; 4Department of Marine Food Science and Technology, Gangneung-Wonju National University, 120 Gangneung Daehangno, Gangneung 210-702, Korea; 5Shanghai Public Health Clinical Center, Shanghai Medical College, Fudan University, Shanghai 201508, China

**Keywords:** *Codium fragile* polysaccharide, mucosal adjuvant, immunotherapy, Lewis lung carcinoma, anti-cancer

## Abstract

Natural polysaccharides have shown promising effects on the regulation of immunity in animals. In this study, we examined the immune stimulatory effect of intranasally administered *Codium fragile* polysaccharides (CFPs) in mice. Intranasal administration of CFPs in C57BL/6 mice induced the upregulation of surface activation marker expression in macrophages and dendritic cells (DCs) in the mediastinal lymph node (mLN) and the production of interleukin-6 (IL-6), IL-12p70, and tumor necrosis factor-α in bronchoalveolar lavage fluid. Moreover, the number of conventional DCs (cDCs) was increased in the mLNs by the upregulation of C-C motif chemokine receptor 7 expression, and subsets of cDCs were also activated following the intranasal administration of CFP. In addition, the intranasal administration of CFPs promoted the activation of natural killer (NK) and T cells in the mLNs, which produce pro-inflammatory cytokines and cytotoxic mediators. Finally, daily administration of CFPs inhibited the infiltration of Lewis lung carcinoma cells into the lungs, and the preventive effect of CFPs on tumor growth required NK and CD8 T cells. Furthermore, CFPs combined with anti-programmed cell death-ligand 1 (PD-L1) antibody (Ab) improved the therapeutic effect of anti-PD-L1 Ab against lung cancer. Therefore, these data demonstrated that the intranasal administration of CFP induced mucosal immunity against lung cancer.

## 1. Introduction

Cancer is the main disease that threatens humankind [1]. Among the various types, lung cancer is characterized by a poor prognosis and difficult diagnosis [2,3]. Most patients with lung cancer are diagnosed at an advanced stage of the disease because there are few symptoms [2,3,4]. For late-stage lung cancer, surgery is not effective, and chemotherapy is associated with undesirable side effects [5,6]. Based on the overexpression of oncoproteins in patients, targeted therapy, such as epidermal growth factor or anaplastic lymphoma kinase, is helpful; however, only a small subgroup of patients are targetable [7]. Recently, various new cancer treatment methods have been developed and applied to lung cancer [7,8,9].

Immunotherapy is a therapeutic method that can selectively kill only the cancer cells by utilizing the toxicity of immune cells, and it is attracting attention because it has fewer side effects and enables efficient cancer treatment [9,10,11,12]. The recent success of therapeutic targeting of the immune checkpoint molecules, especially of programmed cell death 1 (PD-1) and programmed cell death ligand 1 (PD-L1), has changed the paradigm of treatment for several types of cancer [10,12,13,14]. A blockade with PD-1/PD-L1 antibodies (Abs) has thus become a standard-of-care therapeutic option for patients with lung cancer; however, less than 30% of patients respond to the treatment of immune checkpoint blockade therapy [15,16].

For the desired anticancer immunity, cytotoxic immune cell activation is required. Cytotoxic T lymphocytes (CTLs) are immune cell types that mainly contribute to killing cancer cells [17,18]. The induction of CTL activation is mediated by dendritic cells (DCs) [18,19]. DCs are the most potent antigen-presenting cells (APCs), which are activated in response to antigens (Ag) or stimuli [18]. In humans and mice, conventional DCs (cDCs) contain two main populations: cDC1 and cDC2 [20]. cDC1 presents cytosolic Ags to CD8 T cells, which promote Ag-specific CTL activation, whereas cDC2 presents extracellular Ags to CD4 T cells, which express high levels of cytokines for the induction of immune modulation as helper cells [21]. Although the cDCs are functionally divided into two main subsets, activation of both cDC1 and cDC2 is required for eliciting anti-cancer immunity because CTLs cannot be fully activated without helper T cells [20,21,22]. 

Natural killer (NK) cells are another type of immune cell that can contribute to anti-cancer immunity through their cytotoxicity [23]. NK cells that are activated by stimuli upregulate the Fas ligand and tumor necrosis factor (TNF)-related apoptosis-inducing ligand (TRAIL). NK cells expressing these molecules are key effectors for the inhibition of tumor initiation, growth, and metastasis. In addition, NK cells release cytotoxic mediators such as granzyme B and perforin, which induce the apoptosis of cancer cells [24,25,26]. The activation of NK cells is mediated by the stimulation of pattern recognition receptors (PRRs) and interaction with other immune cells, such as DCs and macrophages [27]. The interactions between immune cells are manifested by over-expressed surface proteins and secreted cytokines in the responses to Ags. The interaction with DCs and macrophages in NK cell activation has been extensively studied. In particular, activated cDCs have been shown to induce the production of interferon-gamma (IFN-γ) in NK cells [28]. In addition, DC produces interleukin-12 (IL-12), and interferons (IFNs) elicit the activation of NK cells, which contributes to anticancer immunity [26,29,30]. 

The immunomodulatory effects of natural polysaccharides have been demonstrated in humans and animals [31,32]. Among them, natural marine polysaccharides, such as fucoidans, are well studied for their ability to induce immune activation and anti-cancer immunity. Polysaccharides from *Codium fragile* (CFPs) also show biological activities, including antioxidant, antidiabetic, anti-obesity, and anticancer effects in mice [27,33,34,35]. Moreover, CFPs exhibit an immune-stimulatory effect in humans and mice, which is stronger than that induced by fucoidan. Although the immune stimulatory effect of CFPs has been studied [36,37,38], the effect of CFPs on mucosal immune-cell activation following intranasal administration in mice is yet to be investigated. In this study, we hypothesized that intranasal administration of CFPs can induce mucosal immune cell activation and immunity against lung cancer, and examined the effect of CFPs in mice.

## 2. Results

### 2.1. CFP Induces Activation of DCs and Macrophages in Mediastinal Lymph Nodes (mLNs) 

C57BL/6 mice were intranasally (*i.n*.) injected with CFPs to evaluate their mucosal immune-stimulatory effect, and the mediastinal lymph nodes (mLNs) and BAL fluid were collected (Figure 1A). As shown in Appendix A, the DCs and macrophages were identified in mLNs, and the expression levels of activation markers were measured. Treatment with CFPs dramatically upregulated the expression levels of CD40, 80, 86, and MHC class I and II in both DCs and macrophages, in a dose-dependent manner (Figure 1B and Appendix A). Six hours after the administration of CFPs, the maximum expression levels of the activation markers gradually decreased (Figure 1C and Appendix A). In addition, IL-6, IL-12p70, and tumor necrosis factor-alpha (TNF-α) levels in BAL fluid increased dose-dependently 6 h after CFP administration in mice (Figure 1D). The efficacy of CFPs in activating DCs and macrophages was similar to that of lipopolysaccharide (LPS), a positive control, but the cytokine production levels in BAL fluid were much lower in CFP-administered mice compared to those following treatment with LPS (Figure 1). Therefore, these data indicate that the intranasal administration of CFPs induces the activation of DCs and macrophages.

### 2.2. Subsets of cDCs in mLNs Are Activated by CFPs

Since CTL activation is controlled by cDC1, we next examined whether CFPs can induce the activation of cDC1 and cDC2. CFPs (50 mg/kg) were *i.n.* administered into C57BL/6 mice. Six hours after treatment, the mLNs were harvested, and the subsets of DCs were analyzed. The frequency of cDCs in mLNs was considerably increased by CFP treatment compared to that in the PBS controls (Figure 2A). Consequently, the number of cDCs in the mLNs was significantly increased by CFPs, compared to that in PBS-treated mice (Figure 2B). As is consistent with the increased numbers of cDCs in mLNs, C-C motif chemokine receptor 7 (CCR7) expression levels and the migration factor of activated DCs to lymph nodes were also dramatically upregulated by CFPs (Figure 2C). Compared to PBS-treated control mice, CFP treatment considerably increased the population and frequency of cDC1 in the mLNs (Figure 2D). In addition, both cDC1 and cDC2 upregulated the expression of surface activation markers in response to the intranasal administration of CFPs (Figure 2E). These data indicate that CFPs can induce the activation of both cDC1 and cDC2 in the mLNs. 

### 2.3. NK Cells in the mLNs Are Activated by CFPs

Next, we examined the activation of NK cells by CFPs. C57BL/6 mice were injected intranasally with 50 mg/kg of CFPs. Eighteen hours after injection, mLNs were harvested, and NK cells were identified as shown in Figure 3A. The injection of CFPs promoted an increase in the number of NK cells in the mLNs compared to PBS-treated control (Figure 3B). The surface activation markers in NK cells, including CD69, Fas ligand, and TRAIL, were substantially elevated by CFPs (Figure 3C,D). In addition, intracellular levels of IFN-γ and cytotoxic mediators were also significantly upregulated in mLN NK cells by treatment with CFPs compared to PBS treatment (Figure 3E,F). As shown in Appendix A, the activation markers of NK cells were time-dependently elevated, and rapidly decreased 18 h after treatment with CFPs. Thus, these data suggest that intranasal treatment with CFPs promotes NK cell activation in the mLNs.

### 2.4. CFPs Promoted Production of IFN-γ and TNF-α in T Cells

Our finding that CFPs induce the activation of DCs and macrophages in the mLNs prompted us to examine the effect of CFPs on the induction of T cell differentiation in the mLNs. CFPs were *i.n.* administered to mice twice at three-day intervals. Three days after the last injection, the mice were sacrificed, and the mLNs were harvested. The intracellular IFN-γ and TNF-α levels of CD4 and CD8 T cells were analyzed. CFP treatment remarkably increased the intracellular levels of IFN-γ and TNF-α in both CD4 and CD8 T cells (Figure 4A,B). Although the levels of IFN-γ and TNF-α were significantly increased by CFPs compared to those in the PBS-treated control, these levels were much lower than those in mice following LPS treatment (Figure 4). The levels of intracellular IFN-γ and TNF-α were also increased in a time-dependent manner and reached the maximum on the third day after treatment with CFPs (Appendix A). Thus, these data suggest that CFP treatment through intranasal administration can elicit T cell activation in the mLNs.

### 2.5. Intranasal Administration of CFPs Elicits Immunity against Lewis Lung Carcinoma (LLC) in Mice

Because intranasal administration of CFPs promoted the activation of immune cells in the mLNs, we next examined whether CFPs can elicit immunity against LLC in mice. C57BL/6 mice were injected intravenously (*i.v.*) with LLC cells. Seven days after cell injection, the mice received PBS, 50 mg/kg of CFP, 10 mg/kg of anti-PD-L1 Abs, or a combination of CFPs and anti-PD-L1 Abs. Compared to PBS-treated mice, CFP treatment delayed the death of mice from LLC, a protective effect that was even stronger than that of the anti-PD-L1 Abs (Figure 5A). On day 15 post-tumor injection, LLC cells infiltrated the lungs of PBS-treated mice, whereas CFP administration almost completely prevented LLC cell infiltration in the lungs (Figure 5B,C). Histological analysis indicated that tumor cells were not present in the lungs of mice treated with CFPs, whereas PBS-treated mice showed a large tumor mass in the lungs (Figure 5D). As is consistent with the survival rate data, anti-PD-L1 Ab treatment did not effectively prevent LLC cell infiltration compared to CFPs (Figure 5B–D). Importantly, CFPs when combined with anti-PD-L1 Abs enhanced the protective effect in mice (Figure 5A–D). To evaluate the contribution of CFP-induced immune cells to the anti-cancer effect, NK or CD8 cells were depleted during the treatment of mice. CFP treatment failed to protect against the development of cancer in mice depleted of NK or CD8 cells (Figure 5E), indicating that NK and CD8 cells are required in mice for CFP-mediated protection against LLC. Thus, these data demonstrate that the intranasal administration of CFP elicits anti-cancer immunity and prevents LLC tumor growth in the lungs.

## 3. Discussion

Natural polysaccharides have diverse biochemical and biological properties [13,36,37,39,40]. While many reports have demonstrated the immunosuppressive function of polysaccharides [41,42], only a number of studies have been published on the immune activation function of polysaccharides following in vivo administration [13,33,36,37,39,40]. Moreover, the systemic administration of natural polysaccharides showed limited toxicity in the animal [26,36,43,44]. The nasal administration of drugs is the most effective method for treating lung diseases. In this study, the effect of intranasal administration of CFPs on the activation of immune cells in the lungs and mLNs was examined. Intranasal administration of CFPs induced the activation of DCs, macrophages, T cells, and NK cells in the mLNs, which contributed to anti-cancer responses against LLC cancer growth in the lung. In addition, CTLs and NK cells were demonstrated to be essential for anti-cancer immunity induced by CFPs. Thus, these data demonstrate that the intranasal administration of CFPs elicits immunity against lung cancer in mice. 

Immunotherapeutic approaches for cancer treatment are receiving attention because they can minimize undesirable side effects [9,15]. Activated immune cells secrete cytotoxic substances to kill pathogens, such as bacteria, viruses, and cancer cells [45]. The activation of immune cells that are capable of responding specifically to pathogens is the ultimate direction of immunotherapy [18,46]. To this end, various types of immune stimulators have been developed [11,12,14,47]. In the case of lung cancer, side effects can be minimized by selectively killing only cancer cells in the lungs [2,3,6]. In this regard, it will be desirable to treat lung cancer by administering substances through the nasal cavity [3,6,8]. In this study, we found a novel mucosal immune stimulator, CFP, that promotes the activation of immune cells in the lungs and mLNs. In addition, CFP treatment enhanced the anti-cancer activity of anti-PD-L1 Abs against LLC cells. Therefore, CFPs may be used as a mucosal immune stimulator and adjuvant for immune checkpoint-mediated lung cancer therapy. 

In immunotherapy for cancer, the activation of cytotoxic immune cells is required [9,17,46]. Among the cytotoxic immune cells, CTLs are the most potent immune cells for anti-cancer immunity; they can find cancer cells and directly kill them. The activation of CTLs is controlled by DCs. CD8α expressing cDC1 mainly present Ags through MHC class I and induce CTL activation [9,46]. For the testing of cDC1 activity, cells in the spleen or lymph nodes of the mouse must be analyzed [20,22,48,49,50]. The in vitro differentiated DCs from the bone marrow or monocytes cannot define their subtypes; therefore, the activation of cDC1 cannot be analyzed [48,49]. In this study, we directly analyzed the activation of cDC1 and cDC2 in the mLNs after the administration of CFPs through the nasal cavity. In addition, CFP treatment promoted the accumulation of DCs in the mLNs by overexpressing CCR7. CCR7 is an activation marker and chemokine receptor for homing DCs to the T cell zone in the lymphoid organ [51]. Moreover, the injection of CFPs elicited the release of pro-inflammatory cytokines in the BAL fluid. Taken together, these data demonstrate that the intranasal administration of CFPs induces the maturation of DCs and their differentiation into cDC1 and cDC2 in the mLNs. 

Immune cells interact with and induce the activation of other immune cells [52]. Because innate immune cells highly express PRRs, these cells are preferentially exposed and recognize Ags as well as immune stimulators [53]. DCs and macrophages are phagocytic and Ag-presenting cells, which promote the activation of other cells by the presentation of Ags and by secreting cytokines [53]. In response to immune-stimulatory molecules, DC- and macrophage-secreted cytokines contribute to the differentiation of T cells [54]. It has been shown that these cells produce IL-12, which is the main cytokine that induces the differentiation of naïve T cells into Th1 and Tc1 cells [55,56]. In this study, we found that the IL-12 concentration in the BAL fluid was elevated by the intranasal administration of CFPs, which consequently promoted Th1 and Tc1 differentiation. Tc1 cells are known to be the most important cells for anticancer immunity [18], and the anticancer effect of CFPs was eliminated in mice with CD8 cell depletion. Therefore, these data demonstrate that the intranasal administration of CFPs elicits CTL-mediated immunity against lung cancer. 

NK cells are immune cells that exhibit cytotoxicity against cancer cells [25]. Activated NK cells overexpress CD69, Fas ligand, and TRAIL on their surface, and secrete perforin and granzyme B to kill cancer cells [24,25]. NK cells are activated directly, not only by other stimuli but also through interactions with other immune cells, such as DCs [25]. In a previous study, we demonstrated that the intraperitoneal administration of CFPs can induce NK cell activity [37]. Based on previous data reporting that CFPs can induce the activation of isolated NK cells, it was demonstrated that CFPs can directly activate NK cells [37]. However, as IL-12 can induce NK cell activation, it is suspected that nasally administered CFPs can induce NK cell activity either directly or indirectly [57,58]. Because CFPs induce the production of IL-12, further detailed studies on its direct or indirect action on NK cell activity are needed.

It has been shown that nicotine-mediated tumor progression is developed through the activation of nicotinic acetylcholine receptors (nAChRs), specifically the α7 subunit (α7nAChR) [59,60,61]. The α7nAChRs activate the signaling pathways involved in the proliferation, angiogenesis and metastasis for developing lung cancer [60,62,63]. Therefore, targeting α7nAChR is one of the potential mechanisms that are inevitably the foundation of designing novel anticancer drugs in lung cancer. However, the activation of α7nAChR leads to an anti-inflammatory effect [64], which may contribute to exacerbating the progress of a tumor. Since CFP induces immune cell activation, including DCs, macrophages, T cells and NK cells, it may be possible to elicit immune activation during therapeutic trials by α7nAChR targeting. In the future, we plan to conduct a study on the therapeutic efficacy of CFP and a drug that blocks α7nAChR activity in lung cancer. 

## 4. Materials and Methods

### 4.1. Mice 

Six- to eight-week-old female C57BL/6 mice were purchased from Hyochang Science (Daegu, Korea) and the Shanghai Public Health Clinical Center (SPHCC, Shanghai, China). Mice were housed under specific-pathogen-free conditions at the Laboratory Animal Center of Yeungnam University or SPHCC. All experiments were according to animal-based ethical principles and were conducted according to the IACUC regulations of Yeungnam University (#2021-030) and SPHCC (#2018-A049-01). 

### 4.2. Cancer Cell Line

LLC cell-iRFP-puro (LLC-iRFP-puro) cells were purchased from Imanis life sciences (Rochester, MN, USA). The cells were cultured in RPMI-1640 (Merck KGaA, Darmstadt, Germany) containing 1% penicillin–streptomycin (Thermo Fisher Scientific, Waltham, MA, USA), 2 μg/mL of puromycin (Sigma-Aldrich, St. Louis, MO, USA), and 10% fetal bovine serum (FBS, Sigma-Aldrich, St. Louis, MO, USA) under 5% CO_2_ at 37 °C. 

### 4.3. Preparation of CFPs 

As shown in previous studies, CFPs were extracted from *C. fragile* [38,65]. Briefly, *C. fragile* was incubated with 90% ethanol at 20 °C for 24 h. After the evaporation of ethanol, 65 °C distilled water was added to the samples and incubated for 2 h. The water-soluble sample was harvested, and ethanol was added for filtration. The free proteins in the precipitated samples were removed using the Sevag method after dissolution in distilled water. Using a DEAE Sepharose Fast Flow column (17-0709-01, GE Healthcare Bio-Science AB, Uppsala, Sweden), three fractions (F1, F2, and F3) were separated, and F2 was used as the CFPs. As previously reported [37], the CFPs were mainly composed of carbohydrates (67.4%), proteins (14.7%) and sulfates (10.3%), with a minor amount of uronic acid (2.4%), revealing a purity of over 94.8%.

### 4.4. Reagents and Antibodies

LPS (O111:B4) was obtained from Sigma-Aldrich. Isotype controls and Fc receptor-blocking Abs were obtained from BioLegend (San Diego, CA, USA). FITC-CD3 (17A2, 1:40 dilution with PBS), PerCP5.5-CD4 (GK1.5, 1:50 dilution with PBS), APC/Cyanine7-CD8 (53-5.8, 1:40 dilution with PBS), PerCP/Cyanine5.5-CD11b (M1/70, 1:40 dilution with PBS), ACP/Cyanine7-CD11c (N418, 1:25 dilution with PBS), FITC-CD40 (HM40-3, 1:50 dilution with PBS), ACP/Cyanine7-CD69 (H1.2F3, 1:40 dilution with PBS), Brilliant Violet 605-CD80 (16-10A1, 1:25 dilution with PBS), PE/Cyanine7-CD86 (GL-1, 1:50 dilution with PBS), Alexa Fluor 647-CD197 (C-C chemokine receptor 7;CCR7, 4B12, 1:40 dilution with PBS) PerCP/Cyanine5.5-CD253 (tumor necrosis factor-related apoptosis-inducing ligand; TRAIL, N2B2, 1:25 dilution with PBS), Alexa Flour 647-granzyme B (GB11), PE/Cyanine7-IFN-γ (XMG1.2), PerCP/Cyanine5.5-major histocompatibility complex (MHC) class I (AF6-88.5, 1:40 dilution with PBS), PE-MHC class II (M5/114.15.2, 1:40 dilution with PBS), Brilliant Violet 510-NK1.1 (PK136, 1:40 dilution with PBS), PE-perforin (S16009A) and FITC-TNF-α (MP6-XT22) were also purchased from BioLegend. Anti-PD-L1 Abs were purchased from BioXcell (Lebanon, NH, USA). For the depletion assay of CD8 (YTS169.4) and NK1.1 (PK136)-positive cells, anti-CD8 and anti-NK.1 Abs were obtained from BioXcell. 

### 4.5. Preparation of a Single Cell Suspension of the mLNs

The mLNs were harvested from mice and ground with a glass slide. Aggregated cells and fat were removed using a 100 nm nylon mesh and washed with PBS. The pellets were suspended in 3 mL of histopaque-1077 (Sigma-Aldrich, St. Louis, MO, USA) and the upper layer in 3 mL of fresh histopaque. FBS (1 mL) was top-layered in the cells and the subsequent density by centrifugation at 2000 × *g* for 10 min without breakage. Leukocytes were harvested after washing with PBS. 

### 4.6. Flow Cytometry Analysis

Cells were incubated with Fc receptor-blocking Abs and unlabeled isotype controls at 4 °C for 15 min. The cells were then stained with fluorescence-conjugated Abs on ice for 30 min. After washing with PBS, the cells were resuspended in 4′,6-diamidino-2-phenylindole (DAPI) (Sigma-Aldrich, St. Louis, MO, USA) containing PBS. Multi-color microbeads were prepared as the compensation controls for the analysis of flow cytometry. Auto compensation was used for 9-color staining (NovoExpress, ACEA Biosciences Inc., San Diego, CA, USA). DAPI-negative cells were analyzed using a NovoCyte flow cytometer (ACEA Biosciences Inc., Santa Clara, CA, USA) as live cells.

### 4.7. Analysis of mLN DCs and Macrophages

The mLN cells were stained with anti-CD11b-PE and anti-CD11c-APC/Cyanine7. CD11c^+^CD11b^inter^ cells and CD11c^inter^CD11b^+^ cells were defined as DCs and macrophages, respectively, in the mLNs. The mLN DCs and macrophages were further analyzed for expression levels of CD40, CD80, CD86, and MHC class I and II. 

### 4.8. Analysis of DC Subsets in the mLNs

As described previously [13], the DC subsets in mLN cells were analyzed by flow cytometry. The surface of mLN cells was stained with 9 different color-conjugated Abs. The cells were stained with FITC-conjugated lineage Abs and APC/Cyanine7-conjugated CD11c Abs. The FITC-conjugated lineage Abs included anti-B220, anti-CD3, anti-CD49b, anti-CD90.1, anti-Gr-1, and anti-TER-119. cDC1 and cDC2 were defined as CD8α^+^CD11c^+^lineage^−^ and CD8α^−^CD11c^+^lineage^−^ cells among live leukocytes, respectively. The surface expression levels of CD40, CD80, CD86, MHC class I, and MHC class II in cDC1 and cDC2 were analyzed after intranasal treatment with CFPs. 

### 4.9. NK Cell Analysis

mLNs were harvested from mice after treatment with CFPs or LPS. The mLN cells were stained with fluorescence-conjugated anti-CD3, anti-CD69, anti-Fas ligand, anti-NK1.1, and anti-TRAIL Abs. After staining with DAPI solution (Sigma-Aldrich, St. Louis, MO, USA), CD69, Fas ligand, and TRAIL expression levels in CD3^−^NK1.1^+^DAPI- mLN cells were analyzed to determine NK cell activation. The cells were analyzed using a flow cytometer (ACEA Biosciences Inc., Santa Clara, CA, USA).

### 4.10. Analysis of the Production of Intracellular Cytokines and Cytotoxic Mediators 

After treatment with CFP or LPS, the mLNs from C57BL/6 mice were harvested. The mLN cells were further incubated at 37 °C with Golgistop™ (2 μM monensin solution; BioLegend) for 2 h in vitro. After washing with PBS, the cells were stained using the Zombie Violet Fixable Viability Kit (BioLegend, San Diego, CA, USA) and fixed with fixation buffer (BioLegend, San Diego, CA, USA) at 4 °C for 20 min after staining with surface Abs. The cells were permeabilized using permeabilization wash buffer (BioLegend, San Diego, CA, USA) and intracellular proteins were stained with Abs at 25 °C for 30 min. After washing, the cells were re-suspended in PBS and analyzed using a flow cytometer (ACEA Biosciences Inc., Santa Clara, USA).

### 4.11. Enzyme-Linked Immunosorbent Assay (ELISA) 

Eighteen hours after CFP administration to C57BL/6 mice, bronchoalveolar lavage (BAL) fluid was harvested, and the levels of IFN-γ, IL-6, IL-12p70, and TNF-α were measured using ELISA kits (BioLegend, San Diego, CA, USA). The concentrations of these cytokines were quantified in triplicate.

### 4.12. Mouse LLC Cancer Model and CFP Treatment 

C57BL/6 mice were intravenously administered with LLC-iRFP (0.5 × 10⁶) cells. The tumor-injected mice were randomly divided into PBS, anti-PD-L1 Abs, CFPs, and a combination of anti-PD-L1 Abs and CFPs groups. CFP (50 mg/kg) was intranasally, and anti-PD-L1 Abs (10 mg/kg) was intraperitoneally, administered every 3 days from day 7 after tumor injection. The survival of mice was monitored until day 30 after tumor injection. 

### 4.13. In Vivo Fluorescence Imaging

Metastatic LLC-iRFP cancer in the lungs was imaged using the fluorescence in vivo imaging system, FOBI (Cellgentek, Cheongju, Republic of Korea) on day 15 after the tumor cell challenge of C57BL/6 mice.

### 4.14. Histological Analysis

On day 15 after the tumor challenge, 4% paraformaldehyde (1 mL) was infused into the lungs, then they were harvested. The lungs were then fixed with 4% paraformaldehyde for 24 h and embedded in paraffin. The lungs were sectioned into 5 μm thick slices. After de-paraffinization and rehydration, the lung sections were stained with hematoxylin and eosin (Sigma-Aldrich, St. Louis, MO, USA). 

### 4.15. NK1.1 and CD8-Positive Cell Depletion

NK1.1 or CD8 cells were depleted by the injection of 50 μg anti-NK1.1 (PK136) or anti-CD8α (2.43) Abs (BioXcell, Lebanon, NH, USA) every two days in C57BL/6 mice from day 6 post-tumor injection. The depletion efficacy of NK or CD8 cells was >95%.

### 4.16. Statistical Analysis

Experiments were performed in triplicates with two samples. Data are presented as the mean ± standard error of the mean (SEM). Statistical analysis was performed using ANOVA, followed by Dunnett’s test. Statistical significance was set at *p* < 0.05.

## 5. Conclusions

In conclusion, we found that the intranasal administration of CFPs can elicit immunity against lung cancer in mice. CFPs promoted the activation of macrophages and DC subsets in the mLNs, which further induced the activation of cytotoxic immune cells, including CTLs and NK cells. CFP-induced CTL and NK cell activation prevented LLC tumor growth in the lungs. Moreover, CFPs enhanced the anti-cancer effects of the immune checkpoint blockade. Therefore, it may be possible to induce effective treatment in lung cancer patients whose responses to immune checkpoint inhibitors are insufficient, by concurrently administering the CFP intranasally with immune checkpoint inhibitors. This combination administration method is considered to be a novel lung cancer treatment strategy using CFP.

## Figures and Tables

**Figure 1 ijms-22-10608-f001:**
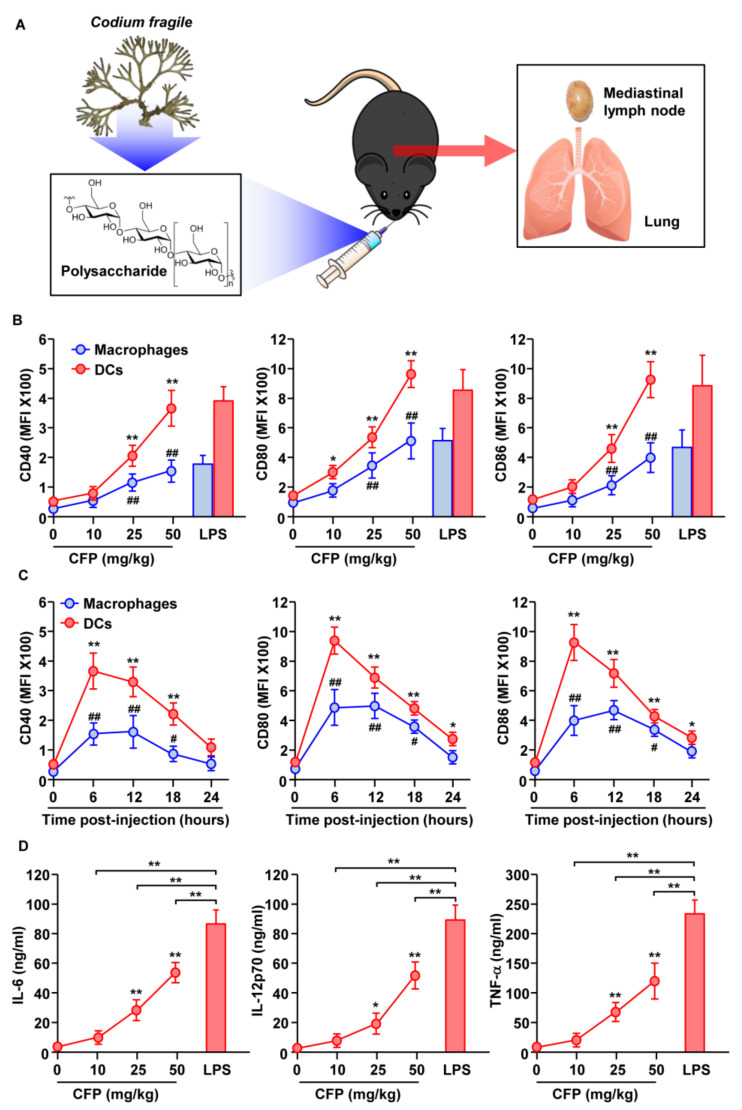
Activation of macrophages and dendritic cells (DCs) in the mediastinal lymph nodes (mLNs) by *Codium fragile* polysaccharides (CFPs). (**A**) C57BL/6 mice were intranasally treated with CFPs and mLN, and bronchoalveolar lavage (BAL) fluid was collected. (**B**) The dose-dependent effect of CFPs on CD40, CD80, and CD86 expression in macrophages and DCs, as analyzed by flow cytometry (*n* = 6 mice, two-way ANOVA, *, *p* < 0.05 vs. 0 mg/kg in DCs; **, *p* < 0.01 vs. 0 mg/kg in DCs; ##, *p* < 0.01 vs. 0 mg/kg in Macrophages). (**C**) The time-dependent effect of CFPs (50 mg/kg) on the expression of costimulatory molecules in macrophages and DCs is shown. (*n* = 6 mice, two-way ANOVA *, *p* < 0.05 vs. 0 h in DCs; **, *p* < 0.01 vs. 0 h in DCs; #, *p* < 0.05 vs. 0 h in Macrophages: ##, *p* < 0.01 vs. 0 h in Macrophages). (**D**) The concentration of interleukin (IL)-6, IL-12p70, and tumor necrosis factor (TNF)-α levels in bronchoalveolar lavage (BAL) fluid, as analyzed by ELISA. (*n* = 6 mice, two-way ANOVA, *, *p* < 0.05 vs. 0 mg/kg; **, *p* < 0.01 vs. 0 mg/kg).

**Figure 2 ijms-22-10608-f002:**
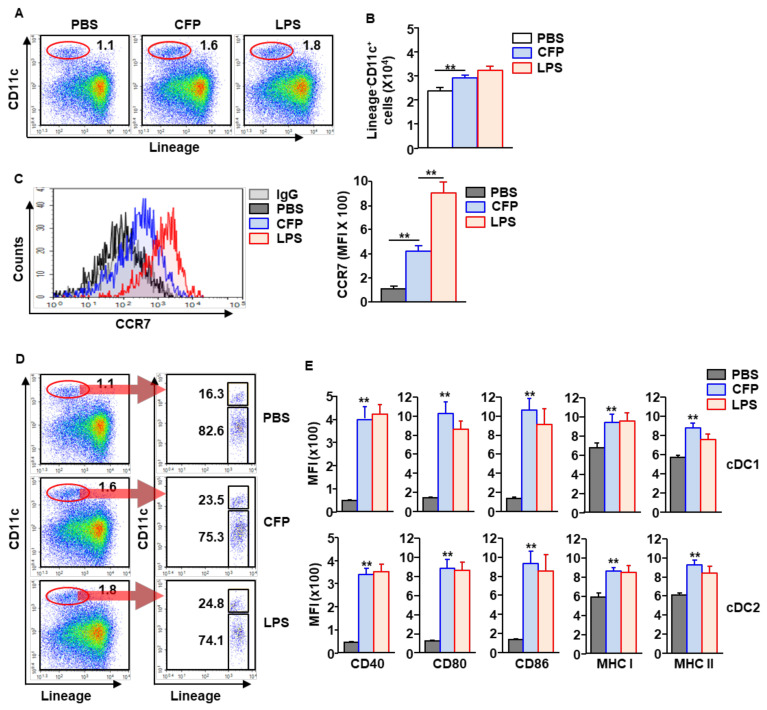
CFPs induce the activation of DC subsets in mLNs. C57BL/6 mice were intranasally administered with 50 mg/kg CFPs. (**A**) Population and frequency of cDCs in the mLNs, analyzed 6 h after treatment. Red circles indicated population of cDCs. (**B**) Absolute numbers of cDCs are shown (*n* = 6 mice, two-way ANOVA, **, *p* < 0.01 vs. PBS treatment). (**C**) Surface expression levels of CCR7 in cDCs were analyzed (left panel). The mean fluorescence intensity (MFI) of CCR7 is shown (right panel, *n* = 6 mice, two-way ANOVA, ** *p* < 0.01). (**D**) Subsets of cDCs, as analyzed by flow cytometry. Red circles indicated population of cDCs. (**E**) MFI of costimulatory and MHC molecules analyzed in the subset of cDCs in the mLNs, 6 h after CFP treatment (*n* = 6 mice, two-way ANOVA, **, *p* < 0.01 vs. PBS treatment).

**Figure 3 ijms-22-10608-f003:**
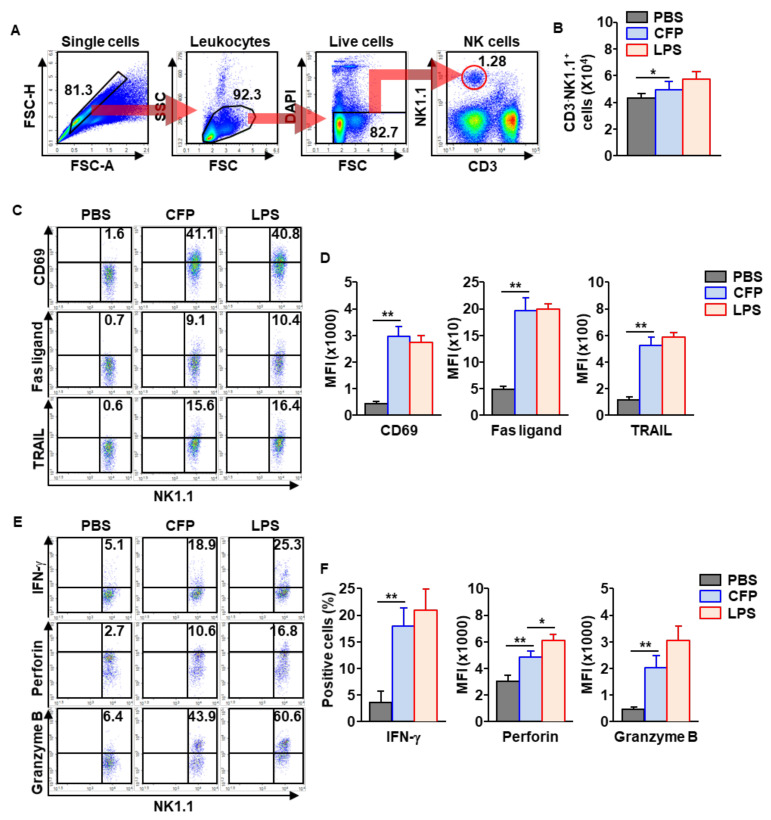
NK cells in the mLNs are activated by CFPs. Eighteen hours after the intranasal administration of 50 mg/kg CFPs, the C57BL/6 mice were sacrificed and the mLNs were harvested. (**A**) The population of NK cells in the mLN is shown. (**B**) The number of NK cells in the mLNs was analyzed (*n* = 6 mice, two-way ANOVA, * *p* < 0.05). (**C**) Surface expression levels of CD69, Fas ligand and tumor necrosis factor-related apoptosis-inducing ligand (TRAIL), measured in mLN NK cells. (**D**) MFI of the indicated molecules is shown (right panel, *n* = 6 mice, two-way ANOVA, ** *p* < 0.01). (**E**) Intracellular levels of IFN-γ, perforin and granzyme B in mLN NK cells, as analyzed. (**F**) The mean positive cells of indicated molecules in NK cells are shown (*n* = 6 mice, two-way ANOVA, * *p* < 0.05, ** *p* < 0.01).

**Figure 4 ijms-22-10608-f004:**
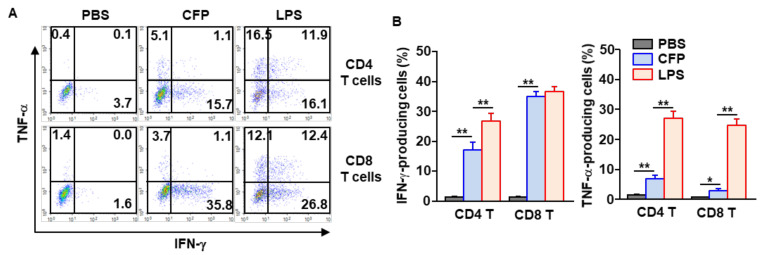
CFPs promote differentiation of T helper 1 and cytotoxic T 1 (Tc1) cells in the mLNs. CFP (50 mg/kg) was intranasally administered to C57BL/6 mice twice at a three days interval. (**A**) The intracellular levels of TNF-α and IFN-γ in CD4 and CD8 T cells in the mLNs. (**B**) Mean percentages of IFN-γ-producing (left panel) and TNF-α-producing (right panel) CD4 and CD8 T cells (*n* = 6, two-way ANOVA, * *p* < 0.05, ** *p* < 0.01).

**Figure 5 ijms-22-10608-f005:**
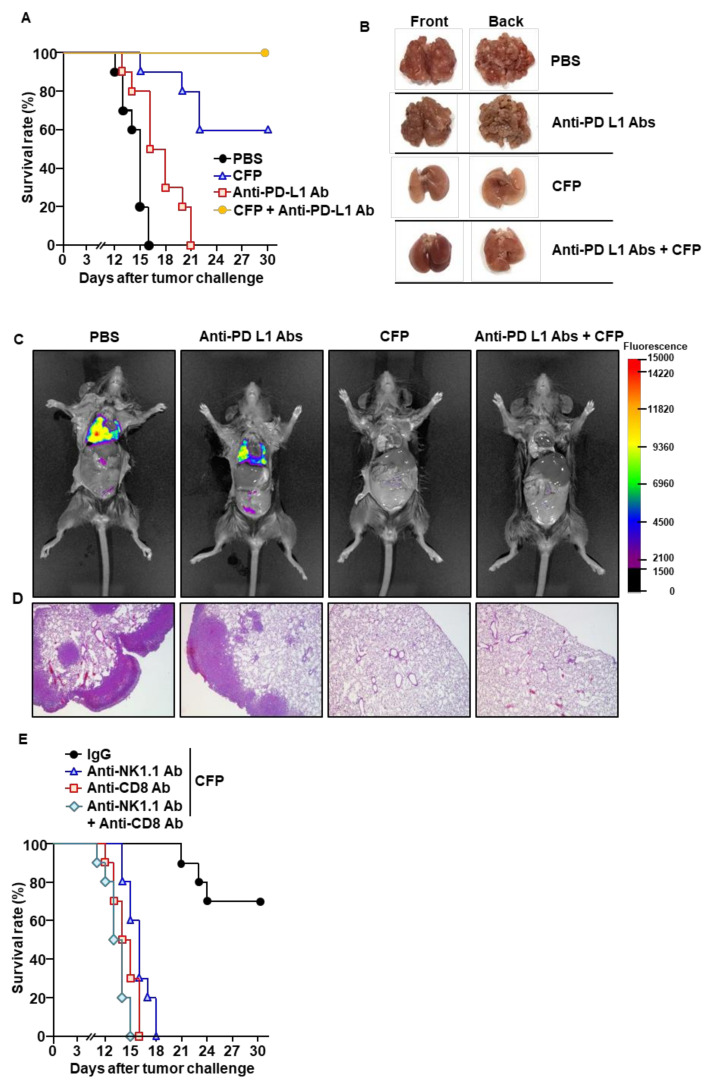
Intranasal administration of CFPs prevents Lewis lung carcinoma (LLC) tumor growth in the lungs. C57BL/6 mice were intravenously (i.v.) administered with LLC-iRFP (0.5 × 10⁶) cells. On day 7 post-tumor injection, the mice were treated with PBS, 50 mg/kg CFP, 10 mg/kg anti-PD-L1 Abs, or a combination of anti-PD-L1 Abs and CFPs. CFPs were intranasally and anti-PD-L1 Abs were intraperitoneally administered to mice every 3 days from day 7 after the tumor injection. (**A**) Survival rates of mice (*n* = 10). (**B**) Lungs, harvested 15 days after the tumor injection. (**C**) Fluorescence image of iRFP, as analyzed. (**D**) LLC cell infiltration in the lungs, analyzed by H&E staining. (**E**) CD8 and NK cells, depleted during treatment with CFPs. The survival rates of mice are shown (*n* = 10).

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
