# Peer review of "Intranasal Administration of Codium fragile Polysaccharide Elicits Anti-Cancer Immunity against Lewis Lung Carcinoma"

_ijms, 2021, doi:10.3390/ijms221910608_

Round 1

Reviewer 1 Report

The article under review is dedicated to the investigation of algal polysaccharide utilization as a possible immunotherapeutic agent against lung cancer. 

The authors demonstrated that polysaccharide application induces maturation of dendritic cells, secretion of cytokines, modulating inflammation by macrophages and dendritic cells as well as by T cells. Finally, they proofed their concept in vivo in LLC model. 

The article is at a high level, well-written, easy to follow. The main conclusions are supported by the results.

Only minor changes can be suggested:

  1. Can you discuss the role of a7-nAChRs in lung cancer progression as a7-nAChR can modulate inflammation (doi: 10.1038/ni1229).
  2. What are the systemic effects of CFP in terms of toxicity and possible side effects? Please discuss.
  3. Please provide dilutions for all Abs used
  4. Please check the necessity for utilization of Tukey test for ANOVA. Its better to use Dunnet test for comparison of multiple values to 1 control. Also, In Figure 1 D its better to use the t-test to compare 50 mg/kg CFP and LPS oe provide ANOVA results for all CFP concentrations.
  5. Generally, the article can be accepted after minor revisions.

Author Response

Reviewer 1

The article under review is dedicated to the investigation of algal polysaccharide utilization as a possible immunotherapeutic agent against lung cancer.

The authors demonstrated that polysaccharide application induces maturation of dendritic cells, secretion of cytokines, modulating inflammation by macrophages and dendritic cells as well as by T cells. Finally, they proofed their concept in vivo in LLC model.

The article is at a high level, well-written, easy to follow. The main conclusions are supported by the results.

Answer (A): We thank for the reviewers comments.

Only minor changes can be suggested:

  1. Can you discuss the role of a7-nAChRs in lung cancer progression as a7-nAChR can modulate inflammation (doi: 10.1038/ni1229).

A: We have added the role of a7-nAChRs and potential trials with CFP for treatment of lung cancer in the ‘discussion’ section.

  1. What are the systemic effects of CFP in terms of toxicity and possible side effects? Please discuss.

A: CFP is a natural polysaccharide derived from seaweed. In previous study, we have examined toxicity of CFP by intraperitoneal administration and the CFP treatment did not induce any toxic effect in the mouse in vivo (doi: 10.3390/md18120626).

  1. Please provide dilutions for all Abs used

A: We have now added to the 'Reagents and antibodies' section of the method.

  1. Please check the necessity for utilization of Tukey test for ANOVA. Its better to use Dunnet test for comparison of multiple values to 1 control. Also, In Figure 1 D its better to use the t-test to compare 50 mg/kg CFP and LPS oe provide ANOVA results for all CFP concentrations.

A: We thank for the important comment for statistical analysis. We have now revised that as the reviewer recommended.

  1. Generally, the article can be accepted after minor revisions.

A: We really appreciate your positive comment. Thanks again.

Reviewer 2 Report

Wang et al. investigated the anti-tumor immune responses induced by Codium fragile polysaccharides (CFPs) after intra-nasal administration in mice. The authors studied the immunostimulatory effects of intranasally administered CFPs on dendritic cells (DCs), macrophages, NKs, and T cells in mediastinal lymph nodes (mLNs) using different markers including cytokines, cell surface markers, and other signaling molecules. They also showed the preclinical efficacy of CFPs in preclinical models of Lewis lung carcinoma after intranasal administration. Overall, the paper is well-written and provides important insights into the immune responses observed after intranasal administration of CFPs in mouse models.

  1. It is known the polysaccharides cause allergenic effects, particularly when inhaled. The authors may need to comment on the potential toxicity of CFPs, if any, in relevant sections of the paper.
  2. It appears that CFPs, as expected, exhibit broad immunostimulatory effects on many immune cell categories. The authors may need to comment on the selectivity of this potential therapeutic strategy and how it may be translated into novel anticancer immunotherapies.
  3. For consistency, I would suggest using the present tense in the titles of “Results” #1-4 as is the case with #5.

Author Response

Wang et al. investigated the anti-tumor immune responses induced by Codium fragile polysaccharides (CFPs) after intra-nasal administration in mice. The authors studied the immunostimulatory effects of intranasally administered CFPs on dendritic cells (DCs), macrophages, NKs, and T cells in mediastinal lymph nodes (mLNs) using different markers including cytokines, cell surface markers, and other signaling molecules. They also showed the preclinical efficacy of CFPs in preclinical models of Lewis lung carcinoma after intranasal administration. Overall, the paper is well-written and provides important insights into the immune responses observed after intranasal administration of CFPs in mouse models.

Answer (A): We thank you for your valuable comments.

1. It is known the polysaccharides cause allergenic effects, particularly when inhaled. The authors may need to comment on the potential toxicity of CFPs, if any, in relevant sections of the paper.

A: Thanks for the comment. As the reviewer mentioned, the polysaccharide may be able to promote allergenic reaction. However, we did not see any allergenic effects of the CFP in the mice as shown in the lung histology data. Due to the allergic immune responses are mediated by Th2, the CFP promoted Th1 may be not contributed to induce the allergenic effects in the lung.

2. It appears that CFPs, as expected, exhibit broad immunostimulatory effects on many immune cell categories. The authors may need to comment on the selectivity of this potential therapeutic strategy and how it may be translated into novel anticancer immunotherapies.

A: In this study, we found that intranasal administration of CFP induced immune activation which further enhanced anti-cancer effect of immune checkpoint inhibitors. As the reviewer well known, the immune checkpoint inhibitors are achieving very impressive results in cancer treatment, but there are many patients who are ineffective, especially lung cancer patients. Therefore, it may be possible to induction of effective treatment to lung cancer patients whose effects of immune checkpoint inhibitors are insufficient by concurrently administering the CFP intranasally with immune checkpoint inhibitors. This combination administration method is considered to be a novel lung cancer treatment strategy using CFP. We have revised the ‘Conclusion’ section.

3.For consistency, I would suggest using the present tense in the titles of “Results” #1-4 as is the case with #5.

A: We have now revised the titles in the “Results” section. 

Reviewer 3 Report

In the present work, the authors have reported on the effects of Codium fragile polysaccharide on the anti-cancer immunity against lung carcinoma. Their approach is very interesting and it has merit for publication. Their work can has merit for publication after addressing the following comments.

I would suggest to add a brief section in the "Introduction" section concerning the cellular pathways of interaction between the immune cells. This would tremendously help the non-specialist understand the activation and interaction pathways between the different immune cell populations. Since the immune system interactions are very complicated, I would suggest to present the populations under investigation in the present work. For example, what is the connection between dendritic cells, macrophages and NK cells? In that way it would be must easier to follow the subsequent investigations concerning those populations.

In the "Materials and Methods" section, please refer to the methodology used for flow cytometry. In particular, how many colors were used? what type and method of color compensation was used?

In addition, please clarify if the plant extracts were given simultaneously to the tumor cells or with time difference.

Finally, please specify which of the four subspecies of the plant were/was used.

In the "Discussion" section please clarify the following: The first comment has to do with the previous methodological concern. That would be, please clarify the time interval with which the plant extracts were administered with respect to the tumor cells. Were they given at the same time? that is on day 1? or tumor cells were first inoculated and the extracts followed?

This is important in terms of drug action. If the the extracts were given simultaneously then they have a protective effect, if later then they have a healing effect. It is very significant and interesting to separate these two phenomena and also comment on the mechanism of action of the plant extract.

Finally the authors should highlight their findings as well as propose future directions for the use of the plant extract in the treatment of lung cancer.

Author Response

In the present work, the authors have reported on the effects of Codium fragile polysaccharide on the anti-cancer immunity against lung carcinoma. Their approach is very interesting and it has merit for publication. Their work can has merit for publication after addressing the following comments.

I would suggest to add a brief section in the "Introduction" section concerning the cellular pathways of interaction between the immune cells. This would tremendously help the non-specialist understand the activation and interaction pathways between the different immune cell populations. Since the immune system interactions are very complicated, I would suggest to present the populations under investigation in the present work. For example, what is the connection between dendritic cells, macrophages and NK cells? In that way it would be must easier to follow the subsequent investigations concerning those populations.

Answer (A): We thank for the reviewers comments. We have now revised the “introduction” section as the reviewer suggested.

In the "Materials and Methods" section, please refer to the methodology used for flow cytometry. In particular, how many colors were used? what type and method of color compensation was used?

A: We have revised the ‘flow cytometry analysis’ section. Moreover, the conjugated fluorescence were also added in the list of antibodies. NovoCyte flow cytometer and NovoExpress software built in compensation method has used. 

In addition, please clarify if the plant extracts were given simultaneously to the tumor cells or with time difference.

A: We are sorry to confuse you. As we mentioned in “methods” and “Figure legends” section (line 202 and 382), CFP (50 mg/kg) was intranasally and anti-PD-L1 Abs (10 mg/kg) were intraperitoneally (i.p.) administered every 3 days from day 7 after tumor injection.

Finally, please specify which of the four subspecies of the plant were/was used.

A: The plant used in this study was Codium fragile aqua-cultured and commercially available in a local market in Wando, Chunnam province, South Korea. The species were identified and confirmed by Biologist Prof. Hyung-Seop Kim, Dept. of Biology, Gangneung-Wonju National University.

In the "Discussion" section please clarify the following: The first comment has to do with the previous methodological concern. That would be, please clarify the time interval with which the plant extracts were administered with respect to the tumor cells. Were they given at the same time? that is on day 1? or tumor cells were first inoculated and the extracts followed?

A: As we mentioned above, on day 7 after tumor injection, the mice were received intranasal administration of CFP for every 3 days. It has been mentioned in the ‘Method’ and ‘Figure legend’.   

This is important in terms of drug action. If the the extracts were given simultaneously then they have a protective effect, if later then they have a healing effect. It is very significant and interesting to separate these two phenomena and also comment on the mechanism of action of the plant extract.

A: Yes, the CFP may have healing effect against the lung cancer, as we shown in the paper. For the prevention effect of CFP, we are also planning to examine. Since the CFP functions as an adjuvant for induction immune cell activation, we will examine combination with CFP and cancer antigen treatment for evaluation prevention against tumor growth.

Finally the authors should highlight their findings as well as propose future directions for the use of the plant extract in the treatment of lung cancer.

A: We have now revised the ‘Conclusion’ section.
